# Investigating medication adherence among Taiwanese patient with hypertension, hyperlipidemia, and diabetes: A pilot study using the Chinese version of a Two-Part Medication Nonadherence Scale and the NHI MediCloud system

**Ya-Wen Lin[1], Pei-Chun Chen[2], Che-Huei Lin[3,4]☯, Ming-Hung Lin👤[3,5]☯ \***

1 School of Nursing, China Medical University, Taichung, Taiwan, R.O.C, 2 Department of Public Health, China Medical University, Taichung, Taiwan, R.O.C, 3 Department of Pharmacy and Master Program, Tajen University, Pingtung, Taiwan, R.O.C, 4 Department of Business and Management, Central Taiwan University of Science and Technology, Taiwan, R.O.C, 5 College of Health, Taichung University of Science and Technology, Taiwan, R.O.C

☯ These authors contributed equally to this work.
\* lmh.roger@msa.hinet.net

**Data Availability Statement:** The data consists of two parts: one is the questionnaire and the other is

## Abstract

### Background

This pilot study aimed to investigate medication nonadherence among Taiwanese patients with diabetes, hypertension, and hyperlipidemia using the Chinese version of the Two-Part Medication Nonadherence Scale (C-TPMNS) and the National Health Insurance (NHI) Medicloud system. The study revealed insights into the factors contributing to nonadherence and the implications for improving patient adherence to medications for chronic conditions. However, the small sample size limits the generalizability of the findings. Additionally, the study identified the need for further research with larger and more diverse samples to validate the preliminary findings.

### Methods

The study conducted surveys individuals in central Taiwan who received three-high medications and those who returned expired medications from chain pharmacies. A structured questionnaire including the C-TPMNS was administered, and additional data on medical history and HbA1c, LDL, and blood pressure levels were collected from the NHI Medicloud system. Data analysis was performed using multiple ordered logistic regression and Wald test methods. Setting interpretation cutoff point to determine medication nonadherence.

### Results

The study found that 25.8% of participants were non-adherent to prescribed medications. Non-adherent individuals had significantly higher systolic blood pressure (SBP ≥ 140

Medical Records. The questionnaire data cannot be shared for sensitive patient information and ethical reasons. For access to the questionnaire data, please contact China Medical University & Hospital Research Ethics Center (Phone:+886-2-2205-2121# 1941; Email: rrec@mail.cmu.edu.tw or irb@mail.cmuh.org.tw) or Miss Zhang Yinxuan (Phone:+886–9-72-500761; Email: y22@ms49.url.com.tw). The medical records database was obtained from the National Health Insurance, and we are not eligible to duplicate and disseminate the database. For further access to the database for medical records in the health insurance MediCloud system, please contact the Ministry of Health and Welfare (Email: stcarolwu@mohw.gov.tw) for further assistance. Taiwan Ministry of Health and Welfare Address: 488 Zhongxiao E. Rd. Sec. 6, Nangang Dist., Taipei 115, Taiwan (R.O.C.). Phone: +886-2-8590-6848.

**Funding:** The author(s) received no specific funding for this work.

**Competing interests:** The authors have declared that no competing interests exist.

**Abbreviations:** MN, Medication nonadherence; MPR, medication possession ratio; C-TPMNS, Two-Part Medication Nonadherence Scale was translated into Chinese; Three highs, Hypertension, Hyperlipidemia, Hyperglycemia; ROC, receiver operating characteristic; TPMNS, Two-Part Medication Nonadherence Scale; NHI Medicloud system, National Health Insurance Medicloud system; SBP, systolic blood pr essure; DBP, diastolic blood pressure; AC, Ante Cibum (before meals); PC, Post Cibum (after meals).

mmHg) than adherent participants. Non-adherence was also associated with factors such as lower education, single status, living alone, abnormal glucose postprandial concentration, and triglyceride levels. The C-TPMNS demonstrated good reliability (Cronbach's alpha = 0.816) and validity (area under the ROC curve = 0.72).

## Conclusion

The study highlighted the complexity of medication nonadherence with diverse determinants and emphasized the importance of tailored interventions. The findings underscored the need for region-specific research to comprehensively address medication nonadherence, especially focusing on adherence to medications for hypertension, hyperlipidemia, and diabetes. The study also identified the need for larger, more diverse studies to validate and expand upon the initial findings and emphasized the importance of pharmacist interventions and patient empowerment in managing chronic conditions and improving overall health outcomes.

## 1. Introduction

The World Health Organization (WHO) defines adherence as "the extent to which the persons' behavior (including medication-taking) corresponds with the agreed recommendations from a healthcare provider" [1] MN is a complex, multifaceted healthcare problem that is associated with the patient, treatment, and/or healthcare provider. If patients do not adhere to treatment, they cannot benefit from their medications, consequently leading to increased mortality and morbidity, as well as healthcare costs [2]. In the United States, approximately 30% to 50% of adults failed to take medications prescribed by healthcare providers resulting in an annual increase in healthcare costs of up to US$100 billion [3] Medication adherence is crucial for diabetes control, impacting mortality, morbidity, and healthcare costs. Globally, non-communicable diseases, especially cardiovascular diseases, pose a significant health challenge [4].

In Taiwan, the three major risk factors for chronic diseases continue to be high blood pressure, high blood sugar, and high cholesterol. Because lifestyle, culture, and health knowledge vary across different countries and regions [5–7], In 2022, heart disease, cerebrovascular diseases, diabetes, and hypertension-related diseases ranked high among causes of death in Taiwan, emphasizing the urgency of addressing health issues, especially diabetes with its 2.27 million diagnosed cases [8]. If not treated properly, diabetes can lead to complications such as retinopathy, nephropathy, neuropathy, and vascular lesions, causing heart attacks, stroke, blindness, and even the need for dialysis or amputation. Patients with diabetes must take long-term medications to control their blood sugar levels but the percentage of medication nonadherence (MN) in patients with diabetes is ≥40%, leading to health deterioration as well as wasting medical resources and medication. Moreover, poor diabetic control can affect the patient's quality of life, thus reducing the MN level among patients with diabetes is crucial. In Taiwan, diabetes and high blood pressure-related diseases are major causes of death [9] Adherent diabetes patients experience reduced hospitalization rates and healthcare costs [10,11] Generally, enhanced medication adherence is linked to reduced hospitalization and ICU admission rates as well as fewer healthcare costs [12]. Using HbA1c level as an indicator, patients with low adherence had a higher level of HbA1c, particularly those prescribed two or

more blood pressure medicines [13], and long-term nonadherence can lead to poor blood sugar control [14], therefore, medication adherence is vital for diabetes control.

The prevalence of MN varies between diseases and ethnic groups, influenced by regimen complexity, age, and medical knowledge [15], Patients with diabetes, hyperlipidemia, and hypertension often face challenges in maintaining adherence due to multiple prescribed medications [16–18] consequently impeding the effort to maintain adherence. A recent study explored the adherence to blood sugar medications among patients in Singapore [19], revealing the younger patients had lower adherence due to long working hours, leading to unsatisfactory blood sugar control as reflected in higher HbA1c levels [20,21]. Another study identified that patients with adequate diabetes knowledge tended to have better medication adherence [22,23] thus education programs on disease and medication management may promote greater disease control [24].

Poor medication adherence (MN), exceeding 40% in diabetes patients, leads to health deterioration, resource wastage, and increased healthcare costs. Taiwan's National Health Insurance system has exposed a worrying trend 80% of outpatient patients use an average of 3.9 medications, of which 25% of these medications going unused, indicating a serious issue of medication wastage [8]. This contributes to rising pharmaceutical expenses and poor disease control, particularly in cases of hypertension, hyperlipidemia, and diabetes. Generally, enhanced medication adherence is linked to reduced hospitalization and ICU admission rates as well as fewer healthcare costs [12]. Using HbA1c level as an indicator, patients with low adherence had a higher level of HbA1c, particularly those prescribed two or more blood pressure medicines [13], and long-term nonadherence can lead to poor blood sugar control [14], therefore, medication adherence is vital for diabetes control.

This study utilizes the Two-Part Medication Nonadherence Scale (TPMNS) to assess MN levels in patients with diabetes, high blood pressure, and high lipids, providing practical suggestions for adherence improvement. The Two-Part Medication Nonadherence Scale (TPMNS) developed by Dr. Corrine Voils was adopted to determine MN [25] levels to offer practical suggestions for adherence improvement in patients with diabetes, high blood pressure, and high lipids. The TPMNS, developed with support from the National Institute on Aging in the United States and a freely available tool, was translated into Chinese and used to explore the MN levels in patients taking medications for diabetes, high blood pressure and high blood lipids [26].

This study, conducted in central Taiwan, aims to understand adherence in patients with high blood pressure, high blood sugar, and high cholesterol obtaining medications from community pharmacies. The other objectives include identifying reasons for medication nonadherence and evaluating the reliability and validity of C-TPMNS [27]. This research provides insights for healthcare professionals and pharmacists to enhance medication adherence, laying the groundwork for tailored health education programs and improvement strategies.

## 2. Methods

### 2.1 Study population and implementation period

This pilot study aimed to investigate medication nonadherence among Taiwanese patients with diabetes, hypertension, and hyperlipidemia using the Chinese version of the Two-Part Medication Nonadherence Scale (C-TPMNS). The study revealed insights into the factors contributing to nonadherence and the implications for improving patient adherence to medications for chronic conditions. The data was collected from individuals who returned expired prescriptions for medications treating high blood pressure, high cholesterol, and diabetes to community pharmacies. The research execution period is from Aug 13, 2020, to Aug. 28, 2021

involving a total of 76 participants. The advantage of using a structured questionnaire is that it allows for the collection of standardized data and compared across participants. It also minimizes the potential for interviewer bias, as all participants are asked the same set of questions in the same way.

## 2.2. Inclusion and exclusion criteria

The patients who visited the community pharmacy for their medicines and those who returned expired medicines. The inclusion criteria were patients with the Diabetes Mellitus, Hypertension, and Hyperlipidemia aged ≥20 years, who are conscious, and had been diagnosed with the three highs at least 6 months before the study commenced. This study conducted surveys among individuals collecting medications for the three highs and those returning expired medicines at a central Taiwan pharmacy chain. Those with speech or cognitive impairment were excluded.

## 2.3 Questionnaire and medical records

**2.3.1 Chinese Two-Part Medication Nonadherence Scale (C-TPMNS).** The TPMNS [28] was employed to evaluate medication nonadherence (MN) and was adapted based on the Chinese version translated by scholars in Singapore. Respondents were also asked to self-report the reasons for their nonadherence. Initially, a pretest was conducted by distributing 30 questionnaires to assess their reliability and validity, and the content was adjusted accordingly. Following this, the formal C-TPMNS questionnaire's reliability and validity were assessed by the research team.

**2.3.2 Medical records in the NHI MediCloud system.** In Taiwan, all patient-prescribed medications are listed in the "MediClound", an online Shared Medication Record accessible by healthcare professionals across sectors, In Taiwan, all prescribed medications for patients are listed in "MediCloud," an online shared medication record accessible to healthcare professionals across departments. The "National Health Insurance Administration (NHIA) Medical Information Cloud Query System" is derived from the National Health Insurance Cloud Medication History System established in 2013. It provides 12 categories of patient medical data for healthcare personnel to access. Since 2018, the system has also implemented mechanisms for medical image sharing, diagnosis, biochemical test data, drug interactions, and allergic medications.

The participants were informed that their data would be collected from the "NHI MediCloud system" and after they consented, the researchers compiled the following information with assistance from pharmacists: medical history, and frequencies of outpatient visits, ICU admission, and hospitalization over the past 3 months. Their physical examination results associated with the three highs(Hypertension, Hyperlipidemia, Hyperglycemia) HbA1c,Low-density lipoprotein, high density lipoprotein, Triglycerides, Blood pressure level over the past 6 months were extracted from the aforementioned system [29].

The medication possession ratio (MPR) for each medication for the "three highs" taken by the participants was recorded and considered a medication adherence indicator, facilitating validity assessment of C-TPMNS. It has been confirmed that the MPR has desirable validity for the use of declaration data of National Health Insurance pharmaceutical products, hence a suitable indicator of medication adherence.MPR refers to the quotient of dividing the number of doses dispensed by healthcare providers by the days of treatment; the ratio can be acquired by using the variable duration denominator (VMPR) or fixed duration denominator. In this study, VMPR was calculated by dividing the sum of days for medication collection by the

number of days between the first prescription and last prescription dates (including supply days).

## 2.4 Data statistics

We used SAS software version 9.4 (SAS Institute Inc., Cary, NC) to perform the data analysis for this study, with a p-value of $< 0.05$ considered statistically significant. The mean and standard deviation were performed correlations between medication adherence and demographic information of the participants. The odds ratio (aOR) with 95% confidence interval (CI) of measured for the correlations between medication adherence and the participants' medical history and medication history obtained from the NHI Medicloud system.

Multivariable logistic regression was used to estimate the odds ratio (aOR) with 95% confidence interval (CI) of each medication history measured for the medication adherence and the medication nonadherence group. The scale validity was assessed according to the MPR value (adherence = MPR>80%) and the area under a receiver operating characteristic (ROC) curve (Fig 1).

## 2.5. Data sources and availability

The data consists of two parts: one is the questionnaire and the other is Medical Records. The The questionnaire data cannot be shared for sensitive patient information and ethical reasons. For access to the questionnaire data, please contact China Medical University & Hospital Research Ethics Center (Phone:+886-2-2205-2121# 1941; Email: rrec@mail.cmu.edu.tw or irb@mail.cmuh.org.tw) or Miss Zhang Yinxuan (Phone:+886—9-72-500761; Email: y22@ms49.url.com.tw). The medical records database were obtained from the National Health Insurance and we are not eligible to duplicate and disseminate the database. For further access to the database for medical records in the health insurance MediCloud system, please contact the Ministry of Health and Welfare (Email: stcarolwu@mohw.gov.tw) for further assistance. Taiwan Ministry of Health and Welfare Address: 488 Zhongxiao E. Rd. Sec. 6, Nangang Dist., Taipei 115, Taiwan (R.O.C.). Phone: +886-2-8590-6848).

## 2.6 Ethics statement

This study confirms that all methods were performed in accordance with the relevant Guidelines and regulations. This study was approved by the Ethics Committee of the Taiwan Taichung China Medical University and Hospital approved the use of data for this study (CMUH- CRREC-108-084(CR-1)). The participants were assured that their answers in this study would not impact their subsequent benefits. Their personal information such as name, address, and identification number were collected to be used in the later follow-up studies, and all data are kept strictly confidential and secured. The data gathered are only used in this study. Investigators read and explained the informed consent, and all participants gave written informed consent before the questionnaire began. All participants willingly provided written informed consent, and the questionnaires were distributed with the assistance of pharmacists and trained personnel. The collected questionnaires were stored and locked in the office to ensure data confidentiality. Only participants ID codes were used throughout the study. Data were not used by a third party. Participants could withdraw at any time without penalty, and their decision would not affect their rights to seek medical treatment. Participation in the study was entirely voluntary, and anonymity was highly guaranteed.

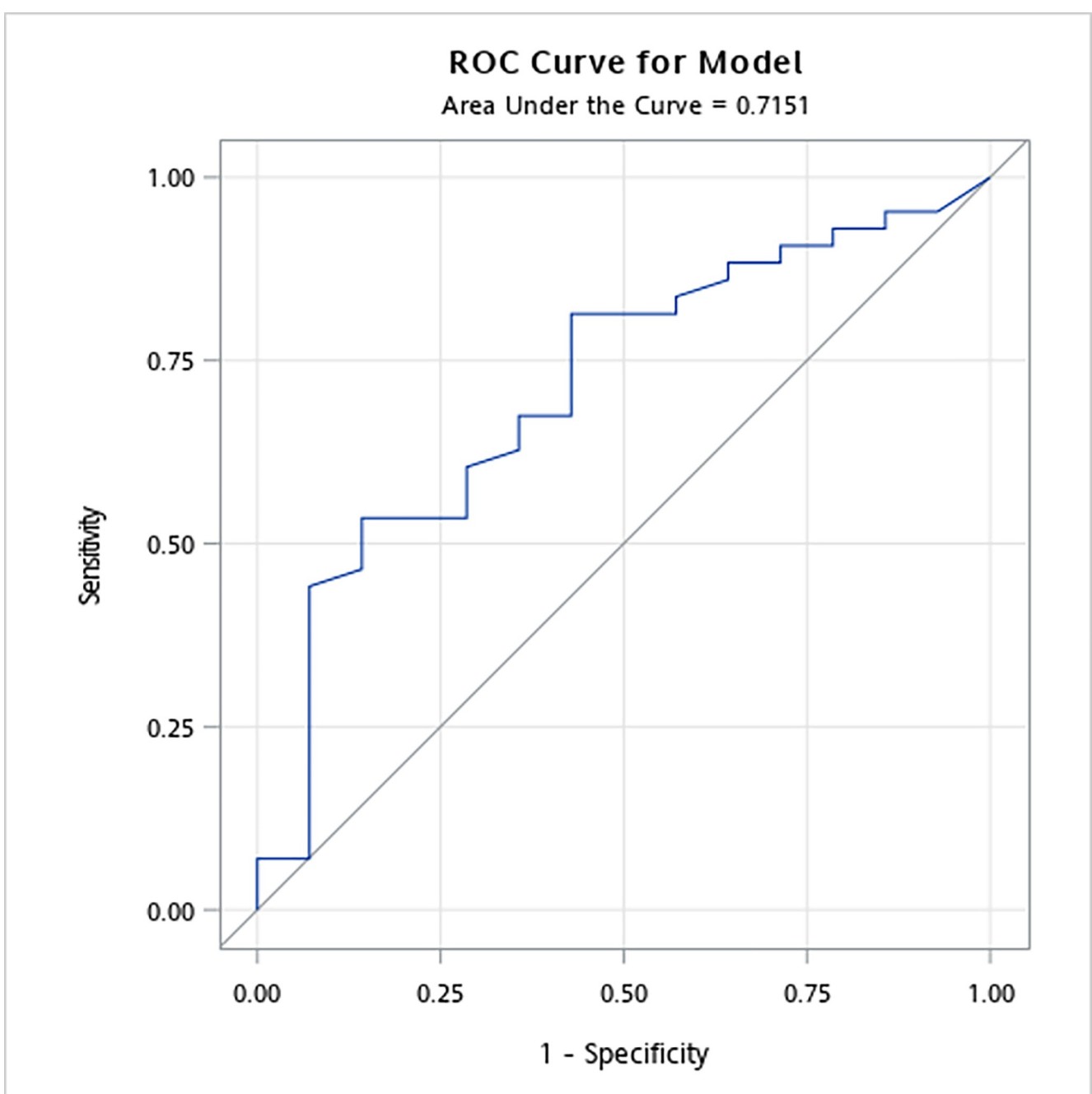

**Fig 1. ROC curve illustrating medication adherence with MPR as the reference.**

## 3 Results

### 3.1 Analysis of pre-test construct validity

The internal consistency reliability of the medication adherence scale was assessed using Cronbach's α, with a value exceeding 0.7 considered indicative of good reliability. Spearman's rank correlation coefficient ρ) value was employed to explore the correlation between the results of the medication adherence scale and Medication Possession Ratio. Additionally, the adherence

scores were dichotomized into adherent and non-adherent groups using different cutoff points, with MPR (>80%) defining adherence. Sensitivity, specificity, positive predictive value, and the area under the receiver operating characteristic curve (AUC) were calculated using MPR as the reference standard, providing an evaluation of construct validity.

## 3.2 The factors responsible for MN in patients with the three highs

Initially, 76 individuals were recruited, out of which seven patients without the three highs and an additional three who completed less than 50% of the questions in the first section (level of MN) of the C-TPMNS (Appendix I) were excluded. Ultimately, this study involved 66 participants, with 49 participants allocated to the adherence group 18 female and 17 to the nonadherence group, resulting in an MN rate of 26%. The patient demographics (Table 1), the majority of patients with the three highs 62.1% were male. Most of patients were employed58.5%. The education level had completed ≥9 years of education were 51.6%, and were either married or cohabitating with their partners (76.6%) or family members (80.3%). While most participants in the adherent group were male, younger, had ≥9 years of education, and were married or cohabitating with their partners and family members, no significant differences were observed between the adherent and no adherent groups. The Wald test for simple logistic regression confirmed the lack of significant correlations between demographic variables and MN (Table 1).

According to the medical records extracted from the NHI Medicloud system, patients without diabetes complications and rheumatoid arthritis exhibited higher adherence rates. Additionally, patients with other diseases tended to display greater adherence compared to those without such diseases. For instance, when compared to patients with depression, those without depression demonstrated a medication-nonadherence odds ratio of 2.67 (95% CI; 0.30–23.42). Similarly, in comparison to patients with kidney disease, those without kidney disease exhibited an odds ratio of 2.23 (95% CI; 0.25–20.02).

To assess the potential impact of regimen complexity on medication adherence, participants were categorized into two groups based on the number of doses per package. Patients taking more than five medicines exhibited a slightly higher nonadherence odds ratio (1.1) compared to those taking four or fewer medicines; however, this difference did not reach statistical significance.

Following the guidelines outlined by the Health Promotion Administration, participants were classified into normal and abnormal groups for each indicator related to the three highs, based on testing results obtained within the past 6 months. Research results show that HbA1c>6.5 accounts for 26%, pre-meal blood sugar (AC) ≥126mg/del is as high as 32.5%. Post-meal blood sugar (PC) ≥200mg/ accounts for 33.3%.The unmoral systolic blood pressure ≥140mmHg and diastolic blood pressure≥90mmHg were 47.1% and 10.2%. The total cholesterol ≥160 mg/dl were 34% and low-density lipoprotein cholesterol ≥100 mg/dl were 11%.

Specifically, Table 2 showed participants with abnormal levels of glucose PC, SBP, DBP, LDL cholesterol, and triglycerides exhibited higher odds of medication nonadherence compared to their counterparts with normal test results. However, only SBP showed a statistically significant correlation with medication nonadherence.

Furthermore, medication adherence was compared among individuals taking different medications for the three highs, based on the resulting Medication Possession Ratios (MPRs). MPRs were 56.6%, 54.6%, and 54% for medications aimed at lowering blood sugar, blood pressure, and blood lipids, respectively (Table 3). Patients displaying high medication adherence (MPR >80%) constituted only 20% of the total participants, with specific percentages of 18.2% for patients taking blood sugar medications, 20.0% for those on blood pressure medications, and 22.6% for patients using medications to manage blood lipids.

**Table 1. Correlations between medication adherence and demographic information of the participants.**

| | Total number of participants | Number of adherent participants ($n = 49$) | Number of nonadherent participants ($n = 17$) | Odds ratio (95%CI) | P value # |
|---|---|---|---|---|---|
| Sex n (%) Men (reference group) Women | 41(62.1%) 25 | 31(63.3%) 18 | 10(58.8%) 7 | 1.00 1.21(0.39–3.72) | 0.75 |
| Age (years) Mean (standard deviation [SD]) (Median, Q1–Q3) | 63.4(14.1) (65,53–74) | 62.2(13.5) (64,53–72) | 66.9(15.5) (69,55–77) | 1.03(0.98–1.07) | 0.23 |
| Height (cm) Mean (SD) (Median, Q1–Q3) | 164.6(7.2) (167,158–170) | 164.1(7.3) (167,158–170) | 165.7(7.2) (167,160–170) | 1.03(0.95–1.12) | 0.44 |
| Weight (kg) Mean (SD) (Median, Q1–Q3) Missing value | 70.9(13.03) (70,63–80) 1 | 69.4(13.1) (70,60–80) 1 | 74.9(12.2) (72,70–80) | 1.03(0.99–1.08) | 0.14 |
| Occupation n (%) | | | | | |
| Employed (reference group) | 38(58.5%) | 28(58.3%) | 10(58.8%) | 1.00 | |
| Retired | 27 | 20 | 7 | 0.98(0.32–3.01) | 0.97 |
| Missing value | 1 | 1 | | | |
| Highest academic level n (%) | | | | | |
| <9 years of education (reference group) | 31(48.4%) | 21(44.7%) | 10(58.8%) | 1.00 | |
| ≥9 years of education | 33 | 26 | 7 | 0.57(0.18–1.74) | 0.32 |
| Missing value | 2 | 2 | | | |
| Marital status n (%) | | | | | |
| Single/divorced/separated/widowed (reference group) | 15(23.4%) | 13(27.7%) | 2(11.8%) | 1.00 | |
| Married/cohabiting | 49 | 34 | 15 | 2.89(0.57–14.32) | 0.20 |
| Missing value | 2 | 2 | | | |
| People they live with n (%) | | | | | |
| Family members (reference group) | 53(80.3%) | 38(77.6%) | 15(88.2%) | 1.00 | |
| Nonfamily members | 13 | 11 | 2 | 0.46(0.09–2.33) | 0.35 |
| Average monthly income n (%) | | | | | |
| ≤NT$30,000 (reference group) | 34(55.7%) | 25(56.8%) | 9(52.9%) | 1.00 | |
| >NT$30,000 | 27 | 19 | 8 | 1.17(0.38–3.60) | 0.78 |
| Missing value | 5 | 5 | | | |

Note: CI = confidence interval.

## 3.3 The reliability and validity of the C-TPMNS

Two experts (a health education scholar who is a native English speaker and a senior pharmacist) were invited to review the first draft of the questionnaire and provide suggestions on its Chinese translation and pharmacy-related content to improve the questionnaire validity. The scale reliability was determined by analyzing the collected responses and was satisfactory (Cronbach's alpha = 0.816). In this study, the Area under the ROC Curve (area under the ROC curve = 0.72) indicates the discriminative ability of the Two-Part Medication Nonadherence Scale (C-TPMNS) in distinguishing between individuals who are adherent to prescribed medications and those who are nonadherent. The reliability and validity of the C-TPMNS were assessed, revealing a moderate predictive power with an AUC of 0.72 in identifying

**Table 2. Correlations between medication adherence and the participants' medical history and medication history obtained from the NHI Medicloud system.**

| | Total number of participants (*n* = 66) | Number of adherent participants (*n* = 49) | Number of nonadherent participants (*n* = 17) | Odds ratio (95%CI) | *P* value # |
|---|---|---|---|---|---|
| **Presence of diabetes *n* (%)** | | | | | |
| Yes (reference group) | 37(56.1%) | 29(59.2%) | 8(47.1%) | 1.00 | 0.39 |
| No | 29 | 20 | 9 | 1.63(0.54–4.95) | |
| **Presence of high blood pressure *n* (%)** | | | | | |
| Yes (reference group) | 56(84.9%) | 41(83.7%) | 15(88.2%) | 1.00 | 0.65 |
| No | 10 | 8 | 2 | 0.68(0.13–3.59) | |
| **Presence of high blood lipids *n* (%)** | | | | | |
| Yes (reference group) | 36(54.6%) | 26(53.1%) | 10(58.8%) | 1.00 | 0.68 |
| No | 30 | 23 | 7 | 0.79(0.26–2.42) | |
| Number of diseases experienced *n* (%) | | | | | |
| One disease (reference group) | 21(31.8%) | 15(30.6%) | 6(35.3%) | 1.00 | |
| Two diseases | 27(40.9%) | 21(42.9%) | 6(35.3%) | 0.71(0.19–2.65) | 0.62 |
| All the three highs | 18(27.3%) | 13(26.5%) | 5(29.4%) | 0.96(0.24–3.90) | 0.96 |
| Medical history *n* (%) | | | | | |
| Presence of medical history | | | | | |
| Yes (reference group) | 55(83.3%) | 40(81.6%) | 15(88.2%) | 1.00 | 0.53 |
| No | 11 | 9 | 2 | 1.69(0.33–8.73) | |
| Presence of diabetes complications | | | | | |
| Yes (reference group) | 20(30.3%) | 13(26.5%) | 7(41.2%) | 1.00 | 0.26 |
| No | 46 | 36 | 10 | 0.52(0.16–1.64) | |
| Presence of depression | | | | | |
| Yes (reference group) | 8(12.1%) | 7(14.3%) | 1(5.9%) | 1.00 | 0.38 |
| No | 58 | 42 | 16 | 2.67(0.30–23.42) | |
| Presence of cancer | | | | | |
| Yes (reference group) | 2(3.03%) | 2(4.1%%) | 0(0.0%) | 1.00 | 0.98 |
| No | 64 | 47 | 17 | >999.99 (<0.001,>999.99) | |
| Presence of kidney disease | | | | | |
| Yes (reference group) | 7(10.6%) | 6(12.2%) | 1(5.9%) | 1.00 | 0.47 |
| No | 59 | 43 | 16 | 2.23(0.25–20.02) | |
| Presence of rheumatoid arthritis | | | | | |
| Yes (reference group) | 7(10.6%) | 5(10.2%) | 2(11.8%) | 1.00 | 0.86 |
| No | 59 | 44 | 15 | 0.85(0.15–4.86) | |
| Presence of other diseases | | | | | |
| Yes (reference group) | 34(51.5%) | 25(51.0%) | 9(52.9%) | 1.00 | 0.89 |
| No | 32 | 24 | 8 | 0.93(0.31–2.80) | |
| Number of doses per package *n* (%) | | | | | |
| ≤4 (reference group) | 27(42.9%) | 20(43.5%) | 7(41.2%) | 1.00 | |
| ≥5 | 36 | 26 | 10 | 1.10(0.36–3.40) | 0.87 |
| Missing value | 3 | 3 | | | |
| Frequencies of hospital visits over the past 3 months (by counts) | | | | | |
| Outpatient visits | | | | | |
| Mean (SD) | 2.7(1.4) | 2.7(1.5) | 2.9(0.8) | 1.15(0.78–1.70) | 0.49 |
| (Median, Q1–Q3) | (3,2–3) | (3,2–3) | (3,3–3) | | |
| Missing value | 3 | 2 | 1 | | |
| ICU admission | | | | | |
| Mean (SD) | 0.1(0.5) | 0.1(0.5) | 0.0(0.0) | 0.006(<0.001,>999.9) | 0.98 |
| (Median, Q1–Q3) | (0,0–0) | (0,0–0) | (0,0–0) | | |
| Missing value | 47 | 33 | 14 | | |
| Hospitalization | | | | | |
| Mean (SD) | 0.3(0.5) | 0.3(0.5) | 0.5(0.6) | 2.80(0.31–25.52) | 0.36 |
| (Median, Q1–Q3) | (0,0–1) | (0,0–1) | (0.5,0–1) | | |
| Missing value | 43 | 30 | 13 | | |

*(Continued)*

**Table 2.** (Continued)

| | Total number of participants (*n* = 66) | Number of adherent participants (*n* = 49) | Number of nonadherent participants (*n* = 17) | Odds ratio (95%CI) | *P* value # |
|---|---|---|---|---|---|
| Testing results of the three highs indicators over the past 6 months | | | | | |
| HbA1c level (%)<br>Mean (SD)<br>(Median, Q1–Q3)<br>Missing value<br>HbA1c level *n* (%)<br><6.5 (reference group)<br>> = 6.5<br>≥6.5<br>Missing value | 6.9(1.3)<br>(6.8,6.2–7.3)<br>26<br>15(37.5%)<br>25<br>26 | 6.9(1.3)<br>(6.8,6.3–7.2)<br>21<br>10(35.7%)<br>18<br>21 | 6.9(1.3)<br>(6.8,5.8–7.6)<br>5<br>5(41.7%)<br>7<br>5 | 0.98(0.58–1.67)<br>1.00<br>0.78(0.20–3.10) | 0.94<br>0.72 |
| Glucose AC (mg/dl)<br>Mean (SD)<br>(Median, Q1–Q3)<br>Missing value<br>Glucose AC (mg/dl) *n* (%)<br><126 (reference group)<br>> = 125<br>≥125<br>Missing value | 121.6(40.2)<br>(118,98–129)<br>26<br>27(67.5%)<br>13<br>26 | 121.2(37.4)<br>(120,100–129)<br>17<br>21(65.6%)<br>11<br>17 | 123.3(53.0)<br>(106.5,91–129)<br>9<br>6(75.0%)<br>2<br>9 | 1.001(0.98–1.02)<br>1.00<br>0.64(0.11–3.70) | 0.89<br>0.61 |
| Glucose PC (mg/dl)<br>Mean (SD)<br>(Median, Q1–Q3)<br>Missing value<br>Glucose PC (mg/dl) *n* (%)<br><200 (reference group)<br>> = 200<br>≥200<br>Missing value | 162.3(58.5)<br>(135,120–202)<br>54<br>8(66.7%)<br>4<br>54 | 135.8(27.9)<br>(124,119.5–140)<br>41<br>7(87.5%)<br>1<br>41 | 215.5(71.3)<br>(214.5,166.5–264.5)<br>13<br>1(25.0%)<br>3<br>13 | 1.04(0.99–1.08)<br>1.00<br>21.00(0.96–458.84) | 0.07<br>0.05 |
| SBP (mmHg)<br>Mean (SD)<br>(Median, Q1–Q3)<br>Missing value<br>SBP (mmHg) *n* (%)<br><140 (reference group)<br>> = 140<br>≥140<br>Missing value | 134.2(16.8)<br>(138,125–148)<br>15<br>27(52.9%)<br>24<br>15 | 131.5(17.3)<br>(138,122–140)<br>12<br>23(62.2%)<br>14<br>12 | 141.5(13.3)<br>(140,138–150)<br>3<br>4(28.6%)<br>10<br>3 | 1.06(0.99–1.12)<br>1.00<br>4.11(1.08–15.63) | 0.07<br>0.04 |
| DBP (mmHg)<br>Mean (SD)<br>(Median, Q1–Q3)<br>Missing value<br>DBP (mmHg) *n* (%)<br><90 (reference group)<br>> = 90<br>≥90<br>Missing value | 78.7(7.1)<br>(80,77–80)<br>17<br>44(89.8%)<br>5<br>17 | 77.7(7.2)<br>(80,71–80)<br>14<br>33(94.3%)<br>2<br>14 | 81.2(6.6)<br>(80,78–80)<br>3<br>11(78.6%)<br>3<br>3 | 1.08(0.98–1.19)<br>1.00<br>4.50(0.66–30.54) | 0.13<br>0.12 |
| Total cholesterol (mg/dl)<br>Mean (SD)<br>(Median, Q1–Q3)<br>Missing value<br>Total cholesterol (mg/dl) *n* (%)<br><160 (reference group)<br>> = 160<br>≥160<br>Missing value | 161.0(40.5)<br>(152.5,131–177)<br>34<br>17(53.1%)<br>15<br>34 | 161.7(43.1)<br>(154,131–173)<br>24<br>13(52.0%)<br>12<br>24 | 158.6(32.2)<br>(151,148–188)<br>10<br>4(57.1%)<br>3<br>10 | 0.99(0.98–1.02)<br>1.00<br>0.81(0.15–4.40) | 0.86<br>0.81 |

(*Continued*)

**Table 2.** (Continued）

| | Total number of participants (*n* = 66) | Number of adherent participants (*n* = 49) | Number of nonadherent participants (*n* = 17) | Odds ratio (95%CI) | *P* value # |
|---|---|---|---|---|---|
| LDL cholesterol (mg/dl)<br>Mean (SD)<br>(Median, Q1–Q3)<br>Missing value<br>LDL cholesterol (mg/dl) *n* (%)<br><100 (reference group)<br>> = 100<br>≥100<br>Missing value | 91.0(26.3)<br>(86,72–106)<br>30<br>25(69.4%)<br>11<br>30 | 89.1(25.1)<br>(85,75–101)<br>20<br>21(72.4%)<br>8<br>20 | 98.7(31.4)<br>(93.5,69–131.6)<br>10<br>4(57.1%)<br>3<br>10 | 1.01(0.98–1.05)<br>1.00<br>1.97(0.36–10.82) | 0.38<br>0.44 |
| Triglycerides (mg/dl)<br>Mean (SD)<br>(Median, Q1–Q3)<br>Missing value<br>Triglycerides (mg/dl) *n* (%)<br><150 (reference group)<br>> = 150<br>≥150 | 146.7(106.9)<br>(113.5,68–200)<br>36<br>19(63.3%)<br>11 | 129.1(72.8)<br>(110,67–191.5)<br>25<br>16(66.7%)<br>8 | 217.0(185.9)<br>(144.5,93–333)<br>11<br>3(50.0%)<br>3 | 1.007(0.99–1.02)<br>1.00<br>2.00(0.33–12.24) | 0.11<br>0.45 |
| Missing value | 36 | 25 | 11 | | |

Note: CI = confident interval.

nonadherence from the collected data (Fig 1). Regarding reasons for medication adherence, the majority of participants reported rarely or never encountering situations that would lead to nonadherence (Table 4).

**Table 3. MPRs (%) of the participants.**

| | (n = 66)<br>Total number of participants (*n* = 66) | (n = 37)<br>Patients taking medications lower blood sugar (*n* = 37) | (n = 56)<br>Patients taking medications lowering blood pressure (*n* = 56) | (n = 36)<br>Patients taking medications lowering blood lipids (*n* = 36) | (n = 21)<br>Patients taking medications for only one of the three highs (*n* = 21) | (n = 27)<br>Patients taking medications for two of the three highs (*n* = 27) | (n = 18)<br>Patients taking medications for all the three highs (*n* = 18) |
|---|---|---|---|---|---|---|---|
| MPR(%)<br>MPR(%)<br>Mean (SD)<br>(Median, Q1–Q3)<br>Missing value | 54.5%(0.2)<br>(48.0%,42.6%-63.6%)<br>9 | 56.0%(0.2)<br>(48.0%,42.6%-63.6%)<br>4 | 54.6%(0.2)<br>(48.0%,42.6%-60.2%)<br>6 | 54.0%(0.3)<br>(44.9%,42.4%-74.3%)<br>5 | 56.8%(0.2)<br>(49.6%,44.8%-67.3%)<br>5 | 50.3%(0.2)<br>(47.7%,42.0%-59.6%)<br>1 | 59.3%(0.3)<br>(44.9%,42.6%-89.6%)<br>3 |
| MPR <0.5 (%)<br>MPR <0.5 (%) | 34(59.7%) | 18(54.5%) | 30(60.0%) | 20(64.5%) | 9(56.2%) | 16(61.5%) | 9(60.0%) |
| 0.5< = MPR<0.8 (%)<br>0.5≤ MPR<0.8 (%) | 13(22.8%) | 9(27.3%) | 10(20.0%) | 4(12.9%) | 5(31.3%) | 6(23.1%) | 2(13.3%) |
| MPR > = 0.8(%)<br>MPR ≥0.8 (%) | 10(17.5%) | 6(18.2%) | 10(20.0%) | 7(22.6%) | 2(12.5%) | 4(15.4%) | 4(26.7%) |

MPR = medicine possession ratio.

**Table 4. Reasons for MN in participants with the three highs.**

| Medication nonadherence reasons | Number of participants (%) | | | | | |
|---|---|---|---|---|---|---|
| | None of the time | A little of the time | Some of the time | Most of the time | Every time | Missing value |
| I was out of my routine | 15(22.7%) | 26(39.4%) | 20(30.3%) | 4(6.1%%) | 1(1.5%) | |
| I forgot | 13(19.7%) | 34(51.5%) | 9(13.6%) | 6(9.1%) | 4(6.1%) | |
| The medication caused side effects. | 22(33.3%) | 24(36.3%) | 17(25.8%) | 1(1.5%) | 2(3.03%) | |
| My irregular eating patterns affected when I took medicines | 23(34.9%) | 24(36.3%) | 13(19.7%) | 5(7.6%) | 1(1.5%) | |
| I did not have my medicines with me | 24(36.3%) | 27(40.9%) | 10(15.1%) | 4(6.1%) | 1(1.5%) | |
| The medication was not working | 24(36.3%) | 29(43.9%) | 11(16.7%) | 2(3.03%) | 0 | |
| I did not want others to see my medications | 22(33.3%) | 30(45.5%) | 8(12.1%) | 3(4.6%) | 3(4.6%) | |
| The medication affected my sex life | 15(23.8%) | 26(41.3%) | 21(33.3%) | 0 | 1(1.6%) | 3 |
| The time for medication was too late | 37(56.1%) | 20(30.3%) | 6(9.1%) | 2(3.03%) | 1(1.5%) | |
| There was no one to help me | 45(69.2%) | 8(12.3%) | 2(3.1%) | 3(4.6%) | 7(10.8%) | 1 |
| Treatment was hard on my family | 44(66.7%) | 9(13.6%) | 3(4.6%) | 9(13.6%) | 1(1.5%) | |
| I had other medications to take | 17(25.8%) | 5(7.6%) | 3(4.6%) | 3(4.6%) | 38(57.6%) | |
| I ran out of medication | 23(35.4%) | 36(55.4%) | 4(6.2%) | 2(3.1%) | 0 | 1 |
| I was afraid the medication would interact with other medications I take | 15(22.7%) | 34(51.5%) | 8(12.1%) | 3(4.6%) | 6(9.1%) | |
| I could not get answers to my questions about the medication | 22(33.9%) | 38(58.5%) | 5(7.7%) | 0 | 0 | 1 |
| I stopped taking the medication when I felt better | 17(25.8%) | 30(45.5%) | 13(19.7%) | 1(1.5%) | 5(7.6%) | |

# 4. Discussion

## 4.1 Challenges in Participant Recruitment processing

This study focused on individuals aged 20 and above who retrieved or disposed of medications at chain pharmacies in the central region. Data were collected through face-to-face surveys. Results revealed that the most commonly discarded medications were those for hypertension, hyperlipidemia, and diabetes. Residents in the central region were most informed about medication disposal through promotional efforts in healthcare institutions. In total, this pilot study recruited 66 participants, of which, 17 (26%) were not adherent to their prescribed medications. The limited sample size is due to the considerable time required for questionnaire completion. Some patients are unwilling or lack the patience to complete the questionnaire fully. As this study relies on voluntary participation, pharmacists cannot compel those returning medications to cooperate as well as those collecting the medications on behalf of the patient. Consequently, their responses might lead to underestimated results of MN, as the respondents might select answers according to social expectations rather than the actual situation, compromising the credibility of their responses. Furthermore, the MN conditions among older adults who live alone, have mobility difficulties, and are cared for by foreign caregivers the population of whom is high in Taiwan cannot be analyzed through a pharmacy-based survey [30,31]. These old people, most prone to age related memory loss, often forget to take medications without others reminding them. Older adults with mobility difficulties might not take their medications regularly due to age-related debility, injuries, and even mild strokes. Despite being looked after by foreign caregivers, those who have experienced a massive stroke or paralysis, or are bedridden as a result of illness might have a low level of medication adherence because of language barriers, cultural differences, neglect, and other reasons.

## 4.2 Factors influencing medication non-adherence patterns

This study revealed that MN is more likely among individuals with lower education levels and those who are single, divorced, separated, or widowed, and do not live with family members. Nonadherent participants had an excessively high level of SBP, glucose PC, and triglycerides, signifying a tendency of poor disease control. However, differences between the adherence and nonadherence groups did not reach significance, except for SBP possibly because of the low sample number, thus lack of statistical power. Taiwan's population is aging, which may be related to increased medication use and medication non-adherence. As in other countries, Taiwanese patients may face barriers to medication adherence such as forgetfulness, medication side effects, and cost [32–35]. In many countries, including Taiwan, social support has been found to be an important factor in medication adherence [36,37]. However, Taiwanese population and those from other Eastern or Western countries in the context of medication cultural factors in Taiwan may play a different role in medication compliance than other countries [38]. For example, traditional Chinese medicine may be used together with western medicine, which may affect medication compliance [39,40].

## 4.3 Unique and healthcare system factors in Taiwan

The healthcare system in Taiwan is different from those in many Western countries, as it is a single-payer system with universal coverage [41,42]. This could affect medication adherence in terms of access to medications and healthcare services. The use of technology in healthcare, such as telemedicine and remote monitoring, may be more prevalent in some Western countries than in Taiwan [43,44], which could affect medication adherence interventions.

Overall, it is important to consider the unique cultural, social, and healthcare system factors that may affect medication adherence in Taiwan and other countries. By understanding these factors, healthcare providers can develop more effective interventions to improve medication adherence and ultimately improve patient [45].

The study highlights the importance of medication adherence in managing chronic conditions such as diabetes, high blood pressure, and high blood lipids. Nonadherence can lead to poor health outcomes, increased healthcare costs, and reduced quality of life for patients and found that forgetfulness, feeling better, and medication side effects were the main factors contributing to medication nonadherence among patients in Taiwan. These findings are consistent with previous research on medication adherence. Tables 1–3 of the study. These tables provide information on the correlations between medication adherence and various demographic, clinical, and medication-related factors. The study also mentions that the Wald test for simple logistic regression was used to confirm that the demographic variables were nonsignificant correlated with medication nonadherence. Overall, the study took steps to control for potential confounding variables and to adjust for differences between groups in the logistic regression analysis.

The study suggests that healthcare providers should take a more patient-centered approach to medication management, which involves understanding the patient's beliefs, values, and preferences regarding their medications [46]. This approach can help healthcare providers tailor their interventions to the patient's specific needs and improve medication adherence and also suggests that healthcare providers should consider using technology-based interventions, such as reminder systems and mobile health apps, to help patients remember to take their medications. These interventions have been shown to be effective in improving medication adherence in previous research.

#### 4.4 Study limitations and considerations for future research

There were several limitations in this study. First, we could not perform the study to compare the adherence of patients with 3highs by community pharmacy or not, because control group without community management was difficulty to be found in Taiwan [47]. Second the study relied on a pharmacy-based survey because it only included individuals who visited pharmacies, which could lead to selection bias. As a pilot study, it was relatively narrow in scope and focused on individuals with specific medical conditions and specific region [48]. Third the cross-sectional design of this study makes it challenging to establish causal relationships between MN and identified factors [49], and the use of self-reported data and potential recall bias may affect the accuracy of the information collected [50]. Fourth the region's cultural and sociodemographic differences may not have been fully represented, limiting the broader applicability of the findings.

### 5 Conclusion

Improving medication compliance in chronic disease patients requires proactive efforts from pharmacists, including regular consultations and education for consistent monitoring of vital indicators. The study confirms the reliability and validity of the C-TPMNS questionnaire, a valuable tool for researchers in similar studies. These interventions, coupled with guidance from healthcare providers, empower patients in managing chronic conditions and improving overall health outcomes.

The study emphasizes the pivotal role of pharmacist consultation and education in enhancing medication compliance, with significant implications for patient outcomes and healthcare costs. Pilot studies are crucial in providing preliminary data and informing larger study designs, laying the groundwork for future research on medication nonadherence in chronic diseases.

Successful government promotion of medication recycling strategies can optimize pharmaceutical resources, reducing health insurance costs and preventing environmental pollution from medication disposal. The severe issue of medication disposal in Taiwan signifies poor adherence among patients, impacting both health control and insurance expenditures. Study participants from central region pharmacies collaborated with pharmacists to access comprehensive medical records through the National Health Insurance system, although the questionnaire process was time-consuming, resulting in a smaller sample. Despite this, our study provides preliminary findings, indicating the importance of future research with a larger sample, especially focusing on adherence to medications for hypertension, hyperlipidemia, and diabetes. This study highlights the collaborative role of health care providers, including doctors and nurses as well as pharmacists, in supporting patients with chronic conditions. Recommendations that pharmacists can implement to improve medication compliance mention the importance of a personalized medication management plan, regular follow-up visits, and effective communication with patients.

### Supporting information

**S1 Appendix. Chinese Two-Part Medication Nonadherence Scale (C-TPMNS).**
(DOCX)

**S1 File.**
(DOC)

## Acknowledgments

The authors would like to thank Dr. Corrine Voils Two-part Medication Nonadherence Scale and This study thanks all study participants for participating.

## Author Contributions

**Conceptualization:** Ya-Wen Lin, Ming-Hung Lin.

**Data curation:** Ya-Wen Lin, Pei-Chun Chen.

**Formal analysis:** Pei-Chun Chen.

**Methodology:** Ya-Wen Lin.

**Supervision:** Che-Huei Lin, Ming-Hung Lin.

**Validation:** Ya-Wen Lin, Che-Huei Lin.

**Writing – original draft:** Che-Huei Lin.

**Writing – review & editing:** Ming-Hung Lin.

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
