## [Decision Letter · Decision Letter 0]

2 Feb 2024

PONE-D-23-38644Investigating Patient Adherence to Medications for Hypertension, Hyperlipidemia, and Diabetes: A Study on Expired Medication Recycling at Community Pharmacies Using the Chinese Version of a Two-Part Medication Nonadherence ScalePLOS ONE

Dear Dr. Lin,

Thank you for submitting your manuscript to PLOS ONE. After careful consideration, we feel that it has merit but does not fully meet PLOS ONE’s publication criteria as it currently stands. Therefore, we invite you to submit a revised version of the manuscript that addresses the points raised during the review process. Please submit your revised manuscript by Mar 18 2024 11:59PM. If you will need more time than this to complete your revisions, please reply to this message or contact the journal office at plosone@plos.org. Please include the following items when submitting your revised manuscript:A rebuttal letter that responds to each point raised by the academic editor and reviewer(s). You should upload this letter as a separate file labeled 'Response to Reviewers'.A marked-up copy of your manuscript that highlights changes made to the original version. You should upload this as a separate file labeled 'Revised Manuscript with Track Changes'.An unmarked version of your revised paper without tracked changes. You should upload this as a separate file labeled 'Manuscript'.

We look forward to receiving your revised manuscript.

Kind regards,

Fadwa Alhalaiqa

Academic Editor

PLOS ONE

Journal Requirements:

**Additional Editor Comments:**

 please consider the following in addition to reviewers' comments: <ul> <li> 

The introduction is too long, require summarization. There are many statements require citation

Method is poorly described, re-write and organize it : research design, inclusion criteria  should be clearly.The results:  re-write the results , the you need to highlight the main findings and the table said the rest.Discussion too many requirements are required see the reviewers comments.Please ensure that your decision is justified on PLOS ONE’s publication criteria and not, for example, on novelty or perceived impact.

Reviewers' comments:

Reviewer's Responses to Questions

**Comments to the Author**

1. Is the manuscript technically sound, and do the data support the conclusions?

Reviewer #1: Partly

Reviewer #2: Yes

2. Has the statistical analysis been performed appropriately and rigorously? 

Reviewer #1: Yes

Reviewer #2: Yes

3. Have the authors made all data underlying the findings in their manuscript fully available?

Reviewer #1: Yes

Reviewer #2: No

4. Is the manuscript presented in an intelligible fashion and written in standard English?

Reviewer #1: No

Reviewer #2: Yes

5. Review Comments to the Author

Reviewer #1: Reviewer # Comment

General comment:

The research topic is interested that addresses the main public health burden that is increasing overtime. However, the sample size is not justified and seems to be too small, so it lacks generalizability. As the authors stated in the limitation part, it is a pilot study, so the phrase “pilot study” better to be included in the title. Lack of generalizability better to be stated under limitation section. Authors are advised to check for grammatical errors and coherence issues.

Specific comments

Abstract

1. Separate what is presented in result and method sections.

2. Data analysis method used, cutoff point for interpretation in method section

3. Write main finding in result section

4. Conclusion should be based on main findings

Introduction

5. Make sure you follow the journal guideline for in text citation.

6. The introduction is too long is there any way to make it precise without losing the main ideas? Check in text citation that some statements not cited.

Method

7. Better to include background information about study setting.

8. Sample size is not justified (too small)

Discussion

9. Is subheading under discussion recommended in the journal guideline?

Reviewer #2: Title: Investigating Patient Adherence to Medications for Hypertension, Hyperlipidemia, and Diabetes: A Study on Expired Medication Recycling at Community Pharmacies Using the Chinese Version of a Two-Part Medication Nonadherence Scale

Abstract:

• Title: the title is not in line with the contents of the article. The title should be revised to reflect the objectives of the study

• ‘’This study conducted surveys among individuals collecting medications for the three highs and those returning expired medicines at a central Taiwan pharmacy chain’’: From this statement the inclusion criteria where patients who visited the community pharmacy for their medicines and those who returned expired medicines. This should be clearly stated in the method section of the manuscript.

Introduction

• Consider breaking down some lengthy sentences for better readability and understanding. For instance, the second paragraph contains several statistics and causes of death. Consider presenting this information in a more concise and structured manner.

• When introducing acronyms like TPMNS, provide a brief explanation the first time they are mentioned.

• Some information is repeated or presented in a slightly different manner. Ensure consistency and eliminate redundancy to maintain a clear and focused narrative.

• If available, include the most recent statistics and data relevant to the context of your study. This ensures that the information is current and aligns with the latest developments.

• Emphasize the significance of your study earlier in the introduction. Clearly state the gap in knowledge or the problem your research aims to address.

• Although adherence is briefly defined, consider providing a clearer and more concise definition of medication nonadherence at the beginning of the introduction.

Method

• The research design should be clearly stated. Questionnaire based approach is not a research design but rather a data collection approach. Information provided under the subheading ‘’Research design and implementation period’’ focuses more on data collection. Kindly revise this section and provide the appropriate information under this section.

• The inclusion criteria should be clearly stated.

• The section on the intervention is very misplaced in the manuscript. Why was the intervention conducted, what was the purpose of conducting the intervention. The objective of the study was to use a tool to assess adherence and then determine factors associated with adherence. So why did the authors conduct an intervention to improve the adherence level of the participants in the study. Are the results presented based on the intervention, if so, is it based on this the baseline data or the end of study data. The authors need to clarify this section.

• The methodology is poorly described. The authors should clearly describe the process of sampling, calculation of sample size, development, and validation of questionnaire, data collection and these should be in line with the objectives of the study.

• Clarify how data from the "NHI MediCloud system" were extracted and linked to individual participants. Ensure that the process of obtaining consent from participants for accessing their data is clearly outlined.

• Explain in more detail the rationale behind choosing the Medication Possession Ratio (MPR) as the indicator for medication adherence. Additionally, provide a brief explanation of VMPR and how it was calculated in this study.

• Provide more information on the specifics of data processing, especially in terms of dealing with missing or incomplete data. Describe any statistical techniques used to address confounding variables or biases.

• While Cronbach’s alpha is mentioned for assessing internal consistency reliability, provide a brief explanation of why this measure was chosen. Additionally, elaborate on the cutoff point used to categorize participants into adherence and nonadherence groups.

• Clearly outline the factors that were considered in the analysis of medication adherence. Provide a rationale for why these specific factors were chosen and discuss any potential limitations in the scope of factors considered.

• Clearly outline the methodology and rationale behind classifying participants into normal and abnormal groups for indicators related to the three highs. Discuss any limitations or considerations in using this classification approach.

• Expand on how the level of medication adherence was compared between different medications. Specify the criteria or metrics used for this comparison and discuss the implications of any observed differences.

• Validation of the C-TPMNS: If not already conducted, perform a validation study of the C-TPMNS against objective measures of medication adherence (e.g., pharmacy refill records, electronic monitoring devices) to confirm its reliability and validity in this population.

• Control for Confounding Variables: Utilize multivariable regression models to control for potential confounding variables that could affect the relationship between the identified factors and MN. This approach would strengthen the causal inferences drawn from the study.

Results

• Specify the exact cutoff points used for dichotomizing adherence scores. Clarify whether multiple cutoff points were considered and provide a rationale for the chosen threshold.

• There results section should provide a brief description of the results and the tables should be referred to. The following information should be provided in a table. ‘’Following the guidelines outlined by the Health Promotion Administration, participants were classified into normal and abnormal groups for each indicator related to the three highs, based on testing results obtained within the past 6 months. An individual was placed in the abnormal group if their HbA1c, glucose ante cibum (AC), glucose post cibum (PC), systolic blood pressure (SBP), diastolic blood pressure (DBP), total cholesterol, LDL-C, low-density lipoprotein (LDL) cholesterol, or triglyceride levels were ≥6.5% (62.5% of total participants) ≥126mg/dl (32.5%), ≥200mg/dl (33.3%), ≥140mmHg (47.1%), ≥90mmHg (10.2%), ≥160mg/dl (46.9%), ≥100mg/dl (30.6%), and ≥150mg/dl (36.7%), respectively’’

• The results section should have less writing and the information. Consider incorporating visual aids, such as tables or figures, to enhance the presentation of complex data, especially when comparing adherence rates or displaying ROC curves.

• Information provided under subheading 3.2 is too lengthy. Authors should modify this section and provide at least two subheadings.

• Table 1: percentages should be stated for the Females

Discussion

• Information provided under subheading 4.1 is not appropriate. Challenges encountered whiles recruiting participants should be placed under limitations of the study. The subheading 4.1 should be modified to reflect the information provided.

.

• Provide more context or possible explanations for the odds ratios presented in relation to diseases like depression and kidney disease. Discuss the clinical significance of these findings and any potential implications for future interventions.

• Elaborate on the concept of regimen complexity and why it was considered as a potential factor influencing medication adherence. Discuss the implications of the observed trend in nonadherence odds ratios for patients taking more than five medicines.

• While Cronbach's alpha and the Area under the ROC Curve are mentioned, briefly explain the significance of these measures in assessing reliability and validity. Additionally, discuss any limitations or potential sources of bias in the validation process.

• While Table 4 mentions that the majority of participants reported rarely or never encountering situations leading to nonadherence, provide more specific information on the reported reasons for medication adherence. This could offer valuable insights into potential intervention targets.

• Expand on the discussion of the unique cultural, social, and healthcare system factors that may influence medication adherence in Taiwan. Provide specific examples or case studies that illustrate how these factors can impact patient behaviour and healthcare delivery.

• Emphasize the importance of a patient-centered approach to medication management and how it can address the specific challenges identified in the study. Discuss potential strategies for healthcare providers to tailor interventions based on patient preferences.

• Provide more details on the types of technology-based interventions that could be effective in improving medication adherence. Discuss the feasibility of implementing such interventions within the Taiwanese healthcare system

Conclusion

• Summarize the key findings of the study in the conclusion section to reinforce the main takeaways for the reader. This can help emphasize the practical implications of the research.

• Strengthen the emphasis on the collaborative role of healthcare providers, including physicians and nurses, in addition to pharmacists, in supporting patients with chronic diseases

• Provide specific recommendations or strategies that pharmacists can implement to improve medication compliance. For example, mention the importance of personalized medication management plans, regular follow-up, and effective communication with patients.

6. PLOS authors have the option to publish the peer review history of their article (what does this mean?). If published, this will include your full peer review and any attached files.

Reviewer #1: No

Reviewer #2: **Yes: **Afia Frimpomaa Asare Marfo

---

## [Author Response · Author response to Decision Letter 0]

1 Apr 2024

Response to Reviewers’ comments

Manuscript ID: [PONE-D-23-38644] - [EMID:ea540dbfb190153a]

MS TITLE: Investigating Patient Adherence to Medications for Hypertension, Hyperlipidemia, and Diabetes: A Study on Expired Medication Recycling at Community Pharmacies Using the Chinese Version of a Two-Part Medication Nonadherence Scale

Response to Editor

Editor Comments to the Author

Reply: 

Thank you for the comments and suggestion. We have revised the manuscript to ensure that our manuscript meets the publication style of PLOS ONE. 

(Please see the revised manuscript.)

Reply: 

Thank you for the editor’s reminder. We have added a description of the supporting information document at the end of the manuscript and updated all text references to match accordingly. 

Reply: 

Thank you for the editor’s reminder. The raw data involved in this analysis are available. The collected questionnaires were stored and locked in the office to ensure data confidentiality. Only participants ID codes were used throughout the study. Data were not used by a third party. Participants could withdraw at any time without penalty, and their decision would not affect their rights to seek medical treatment. Participation in the study was entirely voluntary, and anonymity was highly guaranteed.

Reply: 

Thank you for the reminder. We have added an ethics statement to the Methods section of the manuscript. (Please see the revised manuscript.)

5. Please include captions for your Supporting Information files at the end of your manuscript, and update any in-text citations to match accordingly. Please see our Supporting Information guidelines for more information: http://journals.plos.org/plosone/s/supporting-information

Reply: 

Thank you for the reminder. We have added a description of the supporting information document at the end of the manuscript and updated all text references to match accordingly.

Response to editor;

#Additional Editor Comments:

Please consider the following in addition to reviewers' comments: 

The introduction is too long, require summarization. There are many statements require citation. Method is poorly described, re-write and organize it: research design, inclusion criteria should be clearly. The results: re-write the results, the you need to highlight the main findings and the table said the rest. Discussion too many requirements are required see the reviewers comments. Please ensure that your decision is justified on PLOS ONE’s publication criteria and not, for example, on novelty or perceived impact. For Lab, Study and Registered Report Protocols: These article types are not expected to include results but may include pilot data.

Reply: 

Thank Editor for the comment. We have summarized the introduction again. The method description has been rewritten and organized. The study design has clearly written out the inclusion criteria. The results section has been rewritten with the main findings highlighted and tables representing the rest. The discussion section has been modified to take into account the reviewers' comments. We have re-examined articles for compliance with PLOS ONE's publication criteria rather than factors such as novelty or perceived impact. 

Response to Reviewer 

Reviewer #1 

General comment:

The research topic is interested that addresses the main public health burden that is increasing overtime. However, the sample size is not justified and seems to be too small, so it lacks generalizability. As the authors stated in the limitation part, it is a pilot study, so the phrase “pilot study” better to be included in the title. Lack of generalizability better to be stated under limitation section. Authors are advised to check for grammatical errors and coherence issues.

Specific comments

Abstract

1. Separate what is presented in result and method sections.

2. Data analysis method used, cutoff point for interpretation in method section

3. Write main finding in result section

4. Conclusion should be based on main findings 

Reply:

Thank you for your comments. I have incorporated the suggestions and separated the content based on the specific comments provided. I have also included the phrase "pilot study" in the title and addressed the lack of generalizability under the limitations section. We have modified the title as follow ” Title: Investigating Medication Adherence among Taiwanese Patient with Hypertension, Hyperlipidemia, and Diabetes: A Pilot Study Using the Chinese Version of a Two-Part Medication Nonadherence Scale and the NHI MediCloud system”

Thank you for the suggestion. We have revised the Abstract structured. (Please see below or the revised manuscript.)

Abstract

Background: This pilot study aimed to investigate medication nonadherence among Taiwanese patients with diabetes, hypertension, and hyperlipidemia using the Chinese version of the Two-Part Medication Nonadherence Scale (C-TPMNS) and the National Health Insurance (NHI) MedIcloud system . The study revealed insights into the factors contributing to nonadherence and the implications for improving patient adherence to medications for chronic conditions. However, the small sample size limits the generalizability of the findings. Additionally, the study identified the need for further research with larger and more diverse samples to validate the preliminary findings.

Methods: The study conducted surveys individuals in central Taiwan who received three-high medications and those who returned expired medications from chain pharmacies. A structured questionnaire including the C-TPMNS was administered, and additional data on medical history and HbA1c, LDL, and blood pressure levels were collected from the NHI MedIcloud system. Data analysis was performed using multiple ordered logistic regression and Wald test methods. Setting interpretation cutoff point to determine medication nonadherence.

Results: The study found that 25.8% of participants were non-adherent to prescribed medications. Non-adherent individuals had significantly higher systolic blood pressure (SBP ≥ 140 mmHg) than adherent participants. Non-adherence was also associated with factors such as lower education, single status, living alone, abnormal glucose postprandial concentration, and triglyceride levels. The C-TPMNS demonstrated good reliability (Cronbach's alpha = 0.816) and validity (area under the ROC curve = 0.72).

Conclusion: The study highlighted the complexity of medication nonadherence with diverse determinants and emphasized the importance of tailored interventions. The findings underscored the need for region-specific research to comprehensively address medication nonadherence, especially focusing on adherence to medications for hypertension, hyperlipidemia, and diabetes. The study also identified the need for larger, more diverse studies to validate and expand upon the initial findings and emphasized the importance of pharmacist interventions and patient empowerment in managing chronic conditions and improving overall health outcomes.

Introduction

5. Make sure you follow the journal guideline for in text citation.

6. The introduction is too long is there any way to make it precise without losing the main ideas? Check in text citation that some statements not cited.

Reply:

Thank you for the comment. We have adopted the comment in our revision. We have followed the journal guidelines for in-text citation and made sure that all statements are properly cited. To make the introduction more concise. We have removed some unnecessary details and focused on the main ideas. We have also checked the in-text citation to ensure that all statements are properly cited. We have revised the introduction section. (Please see the revised manuscript.)

Method

7. Better to include background information about study setting.

8. Sample size is not justified (too small)

Reply:

Thank you for the comment. We have adopted the comment in the Method section.

We added background information about the study setting in the Methods section. However, the sample size in this study was small, and there may not be sufficient statistical power to detect weak or moderate differences. The non-compliers were further divided into unintentional non-compliance and intentional non-compliance, and compared with the compliance group to explore the differences in the distribution of various related factors. If the continuous variables were tested, ANOVA was used, and the categorical variables were tested using the Chi-square test, and Further analysis was performed using a multiple ordered logistic regression model, using each relevant factor as the main independent variable to understand the association between each relevant factor and medication noncompliance and its type after mutual correction. (Please see below or the revised manuscript.)

2. Methods

2.1 Research design and implementation period

This pilot study aimed to investigate medication nonadherence among Taiwanese patients with diabetes, hypertension, and hyperlipidemia using the Chinese version of the Two-Part Medication Nonadherence Scale (C-TPMNS). The study revealed insights into the factors contributing to nonadherence and the implications for improving patient adherence to medications for chronic conditions. The data was collected from individuals who returned expired prescriptions for medications treating high blood pressure, high cholesterol, and diabetes to community pharmacies. The research execution period is from Aug 13, 2020, to Aug. 28, 2021. , involving a total of 76 participants. The advantage of using a structured questionnaire is that it allows for the collection of standardized data and compared across participants. It also minimizes the potential for interviewer bias, as all participants are asked the same set of questions in the same way. 

Discussion

9. Is subheading under discussion recommended in the journal guideline?

Reply:

Thank you for the comment. We have revised the discussion section in the journal guideline.

Reviewer #2: 

Title: Investigating Patient Adherence to Medications for Hypertension, Hyperlipidemia, and Diabetes: A Study on Expired Medication Recycling at Community Pharmacies Using the Chinese Version of a Two-Part Medication Nonadherence Scale

Abstract:

• Title: the title is not in line with the contents of the article. The title should be revised to reflect the objectives of the study• ‘’This study conducted surveys among individuals collecting medications for the three highs and those returning expired medicines at a central Taiwan pharmacy chain’’: From this statement the inclusion criteria where patients who visited the community pharmacy for their medicines and those who returned expired medicines. This should be clearly stated in the method section of the manuscript.

Reply:

Thank you for the comment. We have revised the title as follow “Title: Investigating Medication Adherence Among Taiwanese Patient with Hypertension, Hyperlipidemia, and Diabetes: A Pilot Study Using the Chinese Version of a Two-Part Medication Nonadherence Scale and the NHI MediCloud system”. We made it clear in the Methods section of the manuscript that the inclusion criteria were patients who visited community pharmacies to purchase medicines and patients who returned expired medicines. (Please see the revised manuscript.)

Introduction

• Consider breaking down some lengthy sentences for better readability and understanding. For instance, the second paragraph contains several statistics and causes of death. Consider presenting this information in a more concise and structured manner.

• When introducing acronyms like TPMNS, provide a brief explanation the first time they are mentioned.

• Some information is repeated or presented in a slightly different manner. Ensure consistency and eliminate redundancy to maintain a clear and focused narrative.

• If available, include the most recent statistics and data relevant to the context of your study. This ensures that the information is current and aligns with the latest developments.

• Emphasize the significance of your study earlier in the introduction. Clearly state the gap in knowledge or the problem your research aims to address.

• Although adherence is briefly defined, consider providing a clearer and more concise definition of medication nonadherence at the beginning of the introduction.

Reply:

Thank you for the comment. We have adopted the comment in our revision. We have followed the journal guidelines for in-text citation and made sure that all statements are properly cited. To make the introduction more concise. We have removed some unnecessary details and focused on the main ideas. We have also checked the in-text citation to ensure that all statements are properly cited. We have revised the introduction section. (Please see the revised manuscript.)

Method

• The research design should be clearly stated. Questionnaire based approach is not a research design but rather a data collection approach. Information provided under the subheading ‘’Research design and implementation period’’ focuses more on data collection. Kindly revise this section and provide the appropriate information under this section.

• The inclusion criteria should be clearly stated.

• The section on the intervention is very misplaced in the manuscript. Why was the intervention conducted, what was the purpose of conducting the intervention. The objective of the study was to use a tool to assess adherence and then determine factors associated with adherence. So why did the authors conduct an intervention to improve the adherence level of the participants in the study. Are the results presented based on the intervention, if so, is it based on this the baseline data or the end of study data. The authors need to clarify this section.

• The methodology is poorly described. The authors should clearly describe the process of sampling, calculation of sample size, development, and validation of questionnaire, data collection and these should be in line with the objectives of the study.

• Clarify how data from the "NHI MediCloud system" were extracted and linked to individual participants. Ensure that the process of obtaining consent from participants for accessing their data is clearly outlined.

• Explain in more detail the rationale behind choosing the

---

## [Decision Letter · Decision Letter 1]

14 May 2024

Investigating Medication Adherence among Taiwanese Patient with Hypertension, Hyperlipidemia, and Diabetes: A Pilot Study Using the Chinese Version of a Two-Part Medication Nonadherence Scale and the NHI MediCloud system

PONE-D-23-38644R1

Dear Dr. Ming-Hung Lin,

We’re pleased to inform you that your manuscript has been judged scientifically suitable for publication and will be formally accepted for publication once it meets all outstanding technical requirements.

Kind regards,

Fadwa Alhalaiqa

Academic Editor

PLOS ONE

Additional Editor Comments (optional):

Reviewers' comments:

Reviewer's Responses to Questions

**Comments to the Author**

1. If the authors have adequately addressed your comments raised in a previous round of review and you feel that this manuscript is now acceptable for publication, you may indicate that here to bypass the “Comments to the Author” section, enter your conflict of interest statement in the “Confidential to Editor” section, and submit your "Accept" recommendation.

Reviewer #1: All comments have been addressed

2. Is the manuscript technically sound, and do the data support the conclusions?

Reviewer #1: Yes

3. Has the statistical analysis been performed appropriately and rigorously? 

Reviewer #1: Yes

4. Have the authors made all data underlying the findings in their manuscript fully available?

Reviewer #1: Yes

5. Is the manuscript presented in an intelligible fashion and written in standard English?

Reviewer #1: Yes

6. Review Comments to the Author

Reviewer #1: Dear author you have addressed comments raised.

However, still there are some concerns such as in text citation should be in close bracket []. In reference bibliography, if there are more than six authors, list the first six authors et al.,

7. PLOS authors have the option to publish the peer review history of their article (what does this mean?). If published, this will include your full peer review and any attached files.

Reviewer #1: **Yes: **Taklo Simeneh Yazie

---

## [Editor Report · Acceptance letter]

4 Jun 2024

PONE-D-23-38644R1 

PLOS ONE

Dear Dr. Lin, 

I'm pleased to inform you that your manuscript has been deemed suitable for publication in PLOS ONE. Congratulations! Your manuscript is now being handed over to our production team.

Kind regards, 

on behalf of

Pro Fadwa Alhalaiqa 

Academic Editor

PLOS ONE